# Dynamic Channel Pruning: Feature Boosting and Suppression

**Xitong Gao**[1]*, **Yiren Zhao**[2]*, **Łukasz Dudziak**[3], **Robert Mullins**[4], **Cheng-zhong Xu**[5]
[1] Shenzhen Institutes of Advanced Technology, Shenzhen, China
[2,3,4] University of Cambridge, Cambridge, UK
[5] University of Macau, Macau, China
[1] xt.gao@siat.ac.cn, [2] yaz21@cam.ac.uk

## Abstract

Making deep convolutional neural networks more accurate typically comes at the cost of increased computational and memory resources. In this paper, we reduce this cost by exploiting the fact that the importance of features computed by convolutional layers is highly input-dependent, and propose feature boosting and suppression (FBS), a new method to predictively amplify salient convolutional channels and skip unimportant ones at run-time. FBS introduces small auxiliary connections to existing convolutional layers. In contrast to channel pruning methods which permanently remove channels, it preserves the full network structures and accelerates convolution by dynamically skipping unimportant input and output channels. FBS-augmented networks are trained with conventional stochastic gradient descent, making it readily available for many state-of-the-art CNNs. We compare FBS to a range of existing channel pruning and dynamic execution schemes and demonstrate large improvements on ImageNet classification. Experiments show that FBS can respectively provide $5\times$ and $2\times$ savings in compute on VGG-16 and ResNet-18, both with less than 0.6% top-5 accuracy loss.

## 1 Introduction

State-of-the-art vision and image-based tasks such as image classification (Krizhevsky et al., 2012; Simonyan & Zisserman, 2015; He et al., 2016), object detection (Ren et al., 2017; Huang et al., 2017) and segmentation (Long et al., 2015) are all built upon deep *convolutional neural networks* (CNNs). While CNN architectures have evolved to become more efficient, the general trend has been to use larger models with greater memory utilization, bandwidth and compute requirements to achieve higher accuracy. The formidable amount of computational resources used by CNNs present a great challenge in the deployment of CNNs in both cost-sensitive cloud services and low-powered edge computing applications.

One common approach to reduce the memory, bandwidth and compute costs is to prune over-parameterized CNNs. If performed in a coarse-grain manner this approach is known as *channel pruning* (Ye et al., 2018; He et al., 2017; Liu et al., 2017; Wen et al., 2016). Channel pruning evaluates channel saliency measures and removes all input and output connections from unimportant channels— generating a smaller dense model. A saliency-based pruning method, however, has threefold disadvantages. Firstly, by removing channels, the capabilities of CNNs are permanently lost, and the resulting CNN may never regain its accuracy for difficult inputs for which the removed channels were responsible. Secondly, despite the fact that channel pruning may drastically shrink model size, without careful design, computational resources cannot be effectively reduced in a CNN without a detrimental impact on its accuracy. Finally, the saliency of a neuron is not static, which can be illustrated by the feature visualization in Figure 1. Here, a CNN is shown a set of input images, certain channel neurons in a convolutional output may get highly excited, whereas another set of

---

*Equal contribution, corresponding authors.

images elicit little response from the same channels. This is in line with our understanding of CNNs that neurons in a convolutional layer specialize in recognizing distinct features, and the relative importance of a neuron depends heavily on the inputs.

The above shortcomings prompt the question: *why should we prune by static importance, if the importance is highly input-dependent?* Surely, a more promising alternative is to *prune dynamically* depending on the current input. A dynamic channel pruning strategy allows the network to learn to prioritize certain convolutional channels and ignore irrelevant ones. Instead of simply reducing model size at the cost of accuracy with pruning, we can accelerate convolution by selectively computing only a subset of channels predicted to be important at run-time, while considering the sparse input from the preceding convolution layer. In effect, the amount of cached activations and the number of read, write and arithmetic operations used by a well-designed dynamic model can be almost identical to an equivalently sparse statically pruned one. In addition to saving computational resources, a dynamic model preserves all neurons of the full model, which minimizes the impact on task accuracy.

In this paper, we propose *feature boosting and suppression* (FBS) to dynamically amplify and suppress output channels computed by the convolutional layer. Intuitively, we can imagine that the flow of information of each output channel can be amplified or restricted under the control of a "valve". This allows salient information to flow freely while we stop all information from unimportant channels and skip their computation. Unlike pruning statically, the valves use features from the previous layer to predict the saliency of output channels. With conventional *stochastic gradient descent* (SGD) methods, the predictor can learn to adapt itself by observing the input and output features of the convolution operation.

FBS introduces tiny auxiliary connections to existing convolutional layers. The minimal overhead added to the existing model is thus negligible when compared to the potential speed up provided by the dynamic sparsity. Existing dynamic computation strategies in CNNs (Lin et al., 2017; Odena et al., 2017; Bolukbasi et al., 2017) produce on/off pruning decisions or execution path selections. Training them thus often resorts to reinforcement learning, which in practice is often computationally expensive. Even though FBS similarly use non-differentiable functions, contrary to these methods, the unified losses are still well-minimized with conventional SGD.

We apply FBS to a custom CIFAR-10 (Krizhevsky et al., 2014) classifier and popular CNN models such as VGG-16 (Simonyan & Zisserman, 2015) and ResNet-18 (He et al., 2016) trained on the ImageNet dataset (Deng et al., 2009). Empirical results show that under the same speed-ups, FBS can produce models with validation accuracies surpassing all other channel pruning and dynamic conditional execution methods examined in the paper.

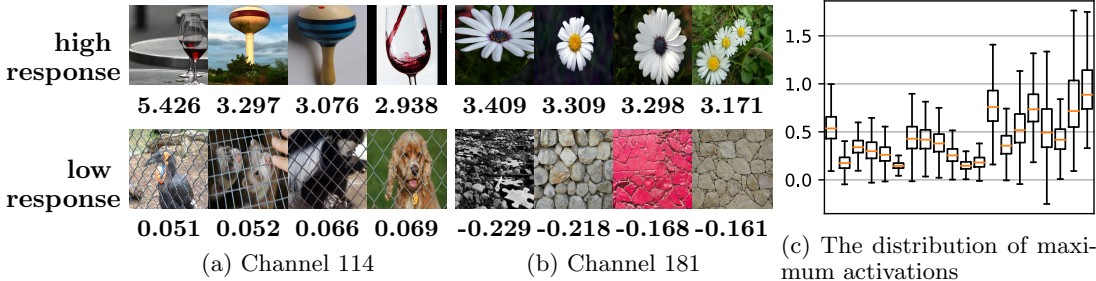

(a) Channel 114    (b) Channel 181    (c) The distribution of maximum activations

Figure 1: When images from the ImageNet validation dataset are shown to a pre-trained ResNet-18 (He et al., 2016), the outputs from certain channel neurons may vary drastically. The top rows in (a) and (b) are found respectively to greatly excite neurons in channels 114 and 181 of layer `block_3b/conv2`, whereas the bottom images elicit little activation from the same channel neurons. The number below each image indicate the maximum values observed in the channel before adding the shortcut and activation. Finally, (c) shows the distribution of maximum activations observed in the first 20 channels.

## 2 Related Work

### 2.1 Structured Sparsity

Since LeCun et al. (1990) introduced optimal brain damage, the idea of creating more compact and efficient CNNs by removing connections or neurons has received significant attention. Early literature on pruning deep CNNs zero out individual weight parameters (Hassibi et al., 1994; Guo et al., 2016). This results in highly irregular sparse connections, which were notoriously difficult for GPUs to exploit. This has prompted custom accelerator solutions that exploit sparse weights (Parashar et al., 2017; Han et al., 2016). Although supporting both sparse and dense convolutions efficiently normally involves some compromises in terms of efficiency or performance.

Alternatively, recent work has thus increasingly focused on introducing *structured sparsity* (Wen et al., 2016; Ye et al., 2018; Alvarez & Salzmann, 2016; Zhou et al., 2016), which can be exploited by GPUs and allows custom accelerators to focus solely on efficient dense operations. Wen et al. (2016) added group Lasso on channel weights to the model's training loss function. This has the effect of reducing the magnitude of channel weights to diminish during training, and remove connections from zeroed-out channels. To facilitate this process, Alvarez & Salzmann (2016) additionally used proximal gradient descent, while Li et al. (2017) and He et al. (2018a) proposed to prune channels by thresholds, *i.e.* they set unimportant channels to zero, and fine-tune the resulting CNN. The objective to induce sparsity in groups of weights may present difficulties for gradient-based methods, given the large number of weights that need to be optimized. A common approach to overcome this is to solve (He et al., 2017) or learn (Liu et al., 2017; Ye et al., 2018) channel saliencies to drive the sparsification of CNNs. He et al. (2017) solved an optimization problem which limits the number of active convolutional channels while minimizing the reconstruction error on the convolutional output. Liu et al. (2017) used Lasso regularization on channel saliencies to induce sparsity and prune channels with a global threshold. Ye et al. (2018) learned to sparsify CNNs with an iterative shrinkage/thresholding algorithm applied to the scaling factors in batch normalization. There are methods (Luo et al., 2017; Zhuang et al., 2018) that use greedy algorithms for channel selection. Huang et al. (2018) and He et al. (2018b) adopted reinforcement learning to train agents to produce channel pruning decisions. PerforatedCNNs, proposed by Figurnov et al. (2016), use predefined masks that are model-agnostic to skip the output pixels in convolutional layers.

### 2.2 Dynamic Execution

In a pruned model produced by structured sparsity methods, the capabilities of the pruned neurons and connections are permanently lost. Therefore, many propose to use dynamic networks as an alternative to structured sparsity. During inference, a dynamic network can use the input data to choose parts of the network to evaluate.

Convolutional layers are usually spatially sparse, *i.e.* their activation outputs may contain only small patches of salient regions. A number of recent publications exploit this for acceleration. Dong et al. (2017) introduced low-cost collaborative layers which induce spatial sparsity in cheap convolutions, so that the main expensive ones can use the same sparsity information. Figurnov et al. (2017) proposed spatially adaptive computation time for residual networks (He et al., 2016), which learns the number of residual blocks required to compute a certain spatial location. Almahairi et al. (2016) presented dynamic capacity networks, which use the gradient of a coarse output's entropy to select salient locations in the input image for refinement. Ren et al. (2018) assumed the availability of *a priori* spatial sparsity in the input image, and accelerated the convolutional layer by computing non-sparse regions.

There are dynamic networks that make binary decisions or multiple choices for the inference paths taken. BlockDrop, proposed by Wu et al. (2018), trains a policy network to skip blocks in residual networks. Liu & Deng (2018) proposed conditional branches in deep neural networks (DNNs), and used Q-learning to train the branching policies. Odena et al. (2017) designed a DNN with layers containing multiple modules, and decided which module to use with a recurrent neural network (RNN). Lin et al. (2017) learned an RNN to

adaptively prune channels in convolutional layers. The on/off decisions commonly used in these networks cannot be represented by differentiable functions, hence the gradients are not well-defined. Consequently, the dynamic networks above train their policy functions by reinforcement learning. There exist, however, methods that workaround such limitations. Shazeer et al. (2017) introduced sparsely-gated mixture-of-experts and used a noisy ranking on the backpropagate-able gating networks to select the expensive experts to evaluate. Bolukbasi et al. (2017) trained differentiable policy functions to implement early exits in a DNN. Hua et al. (2018) learned binary policies that decide whether partial or all input channels are used for convolution, but approximate the gradients of the non-differentiable policy functions with continuous ones.

## 3 Feature Boosting and Suppression

We start with a high-level illustration (Figure 2) of how FBS accelerates a convolutional layer with *batch normalization* (BN). The auxiliary components (in red) predict the importance of each output channel based on the input features, and amplify the output features accordingly. Moreover, certain output channels are predicted to be entirely suppressed (or zero-valued as represented by ⊘), such output sparsity information can advise the convolution operation to skip the computation of these channels, as indicated by the dashed arrow. It is notable that the expensive convolution can be doubly accelerated by skipping the inactive channels from both the input features and the predicted output channel saliencies. The rest of this section provides detailed explanation of the components in Figure 2.

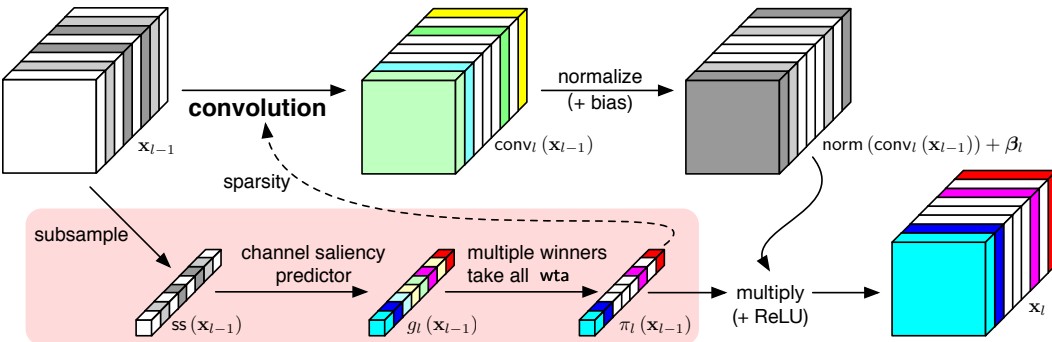

Figure 2: A high level view of a convolutional layer with FBS. By way of illustration, we use the $l^{\text{th}}$ layer with 8-channel input and output features, where channels are colored to indicate different saliencies, and the white blocks (⊘) represent all-zero channels.

### 3.1 Preliminaries

For simplicity, we consider a deep sequential batch-normalized (Ioffe & Szegedy, 2015) CNN with $L$ convolutional layers, *i.e.* $\mathbf{x}_L = F(\mathbf{x}_0) = f_L(\cdots f_2(f_1(\mathbf{x}_0))\cdots)$, where the $l^{\text{th}}$ layer $f_l : \mathbb{R}^{C_{l-1} \times H_{l-1} \times W_{l-1}} \to \mathbb{R}^{C_l \times H_l \times W_l}$ computes the features $\mathbf{x}_l \in \mathbb{R}^{C_l \times H_l \times W_l}$, which comprise of $C_l$ channels of features with height $H_l$ and width $W_l$. The $l^{\text{th}}$ layer is thus defined as:

$$f_l(\mathbf{x}_{l-1}) = (\boldsymbol{\gamma}_l \cdot \text{norm}(\text{conv}_l(\mathbf{x}_{l-1}, \boldsymbol{\theta}_l)) + \boldsymbol{\beta}_l)_+. \tag{1}$$

Here, additions ($+$) and multiplications ($\cdot$) are element-wise, $(\mathbf{z})_+ = \max(\mathbf{z}, 0)$ denotes the ReLU activation, $\boldsymbol{\gamma}_l, \boldsymbol{\beta}_l \in \mathbb{R}^{C_l}$ are trainable parameters, $\text{norm}(\mathbf{z})$ normalizes each channel of features $\mathbf{z}$ across the population of $\mathbf{z}$, with $\boldsymbol{\mu}_{\mathbf{z}}, \boldsymbol{\sigma}_{\mathbf{z}}^2 \in \mathbb{R}^{C_l}$ respectively containing the population mean and variance of each channel, and a small $\epsilon$ prevents division by zero:

$$\text{norm}(\mathbf{z}) = \frac{\mathbf{z} - \boldsymbol{\mu}_{\mathbf{z}}}{\sqrt{\boldsymbol{\sigma}_{\mathbf{z}}^2 + \epsilon}}. \tag{2}$$

Additionally, $\text{conv}_l(\mathbf{x}_{l-1}, \boldsymbol{\theta}_l)$ computes the convolution of input features $x_{l-1}$ using the weight tensor $\boldsymbol{\theta}_l \in \mathbb{R}^{C^l \times C^{l-1} \times k^2}$, where $k$ is the kernel size. Specifically, FBS concerns the

optimization of $\mathsf{conv}_l\left(\mathbf{x}_{l-1}, \boldsymbol{\theta}_l\right)$ functions, as a CNN spends the majority of its inference time in them, using $k^2 C_{l-1} C_l H_l W_l$ *multiply-accumulate operations* (MACs) for the $l^{\text{th}}$ layer.

## 3.2 Designing a Dynamic Layer

Consider the following generalization of a layer with dynamic execution:

$$\hat{f}\left(\mathbf{x}, \cdots\right) = f\left(\mathbf{x}, \boldsymbol{\theta}, \cdots\right) \cdot \pi\left(\mathbf{x}, \boldsymbol{\phi}, \cdots\right), \tag{3}$$

where $f$ and $\pi$ respectively use weight parameters $\boldsymbol{\theta}$ and $\boldsymbol{\phi}$ and may have additional inputs, and compute tensors of the same output shape, denoted by $\mathbf{F}$ and $\mathbf{G}$. Intuitively, the expensive $\mathbf{F}^{[\mathbf{i}]}$ can always be skipped for any index $\mathbf{i}$ whenever the cost-effective $\mathbf{G}^{[\mathbf{i}]}$ evaluates to $\mathbf{0}$. Here, the superscript $[\mathbf{i}]$ is used to index the $\mathbf{i}^{\text{th}}$ slice of the tensor. For example, if we have features $\mathbf{F} \in \mathbb{R}^{C \times H \times W}$ containing $C$ channels of $H$-by-$W$ features, $\mathbf{F}^{[c]} \in \mathbb{R}^{H \times W}$ retrieves the $c^{\text{th}}$ feature image. We can further sparsify and accelerate the layer by adding, for instance, a Lasso on $\pi$ to the total loss, where $\mathbb{E}_{\mathbf{x}}\left[\mathbf{z}\right]$ is the expectation of $\mathbf{z}$ over $\mathbf{x}$:

$$\mathcal{R}\left(\mathbf{x}\right) = \mathbb{E}_{\mathbf{x}}\left[\left\|\pi\left(\mathbf{x}, \boldsymbol{\phi}, \cdots\right)\right\|_1\right], \tag{4}$$

Despite the simplicity of this formulation, it is however very tricky to design $\hat{f}$ properly. Under the right conditions, we can arbitrarily minimize the Lasso while maintaining the same output from the layer by scaling parameters. For example, in low-cost collaborative layers (Dong et al., 2017), $f$ and $\pi$ are simply convolutions (with or without ReLU activation) that respectively have weights $\boldsymbol{\theta}$ and $\boldsymbol{\phi}$. Since $f$ and $\pi$ are homogeneous functions, one can always halve $\boldsymbol{\phi}$ and double $\boldsymbol{\theta}$ to decrease (4) while the network output remains the same. In other words, the optimal network must have $\left\|\boldsymbol{\phi}\right\|_\infty \to 0$, which is infeasible in finite-precision arithmetic. For the above reasons, Dong et al. (2017) observed that the additional loss in (4) always degrades the CNN's task performance. Ye et al. (2018) pointed out that gradient-based training algorithms are highly inefficient in exploring such reparameterization patterns, and channel pruning methods may experience similar difficulties. Shazeer et al. (2017) avoided this limitation by finishing $\pi$ with a softmax normalization, but (4) can no longer be used as the softmax renders the $\ell^1$-norm, which now evaluates to 1, useless. In addition, similar to sigmoid, softmax (without the cross entropy) is easily saturated, and thus may equally suffer from vanishing gradients. Many instead design $\pi$ to produce on/off decisions and train them with reinforcement learning as discussed in Section 2.

## 3.3 Feature Boosting and Suppression with Channel Saliencies

Instead of imposing sparsity on features or convolutional weight parameters (*e.g.* Wen et al. (2016); Alvarez & Salzmann (2016); Li et al. (2017); He et al. (2018a)), recent channel pruning methods (Liu et al., 2017; Ye et al., 2018) induce sparsity on the BN scaling factors $\boldsymbol{\gamma}_l$. Inspired by them, FBS similarly generates a channel-wise importance measure. Yet contrary to them, instead of using the constant BN scaling factors $\boldsymbol{\gamma}_l$, we predict channel importance and dynamically amplify or suppress channels with a parametric function $\pi(\mathbf{x}_{l-1})$ dependent on the output from the previous layer $\mathbf{x}_{l-1}$. Here, we propose to replace the layer definition $f_l\left(\mathbf{x}_{l-1}\right)$ for each of $l \in [1, L]$ with $\hat{f}_l\left(\mathbf{x}_{l-1}\right)$ which employs dynamic channel pruning:

$$\hat{f}_l\left(\mathbf{x}_{l-1}\right) = \left(\pi_l\left(\mathbf{x}_{l-1}\right) \cdot \left(\mathsf{norm}\left(\mathsf{conv}_l\left(\mathbf{x}_{l-1}, \boldsymbol{\theta}_l\right)\right) + \boldsymbol{\beta}_l\right)\right)_+, \tag{5}$$

where a low-overhead policy $\pi_l\left(\mathbf{x}_{l-1}\right)$ evaluates the pruning decisions for the computationally demanding $\mathsf{conv}\left(\mathbf{x}_{l-1}, \boldsymbol{\theta}_l\right)$:

$$\pi_l\left(\mathbf{x}_{l-1}\right) = \mathsf{wta}_{\lceil dC_l \rceil}\left(g_l\left(\mathbf{x}_{l-1}\right)\right). \tag{6}$$

Here, $\mathsf{wta}_k(\mathbf{z})$ is a $k$-winners-take-all function, *i.e.* it returns a tensor identical to $\mathbf{z}$, except that we zero out entries in $\mathbf{z}$ that are smaller than the $k$ largest entries in absolute magnitude. In other words, $\mathsf{wta}_{\lceil dC_l \rceil}(g_l(\mathbf{x}_{l-1}))$ provides a pruning strategy that computes only $\lceil dC_l \rceil$ most salient channels predicted by $g_l(\mathbf{x}_{l-1})$, and suppresses the remaining channels with zeros. In Section 3.4, we provide a detailed explanation of how we design a cheap $g_l(\mathbf{x}_{l-1})$ that learns to predict channel saliencies.

It is notable that our strategy prunes $C_l - \lceil dC_l \rceil$ least salient output channels from $l^{\text{th}}$ layer, where the density $d \in \,]0, 1]$ can be varied to sweep the trade-off relationship between performance and accuracy. Moreover, pruned channels contain all-zero values. This allows the subsequent $(l + 1)^{\text{th}}$ layer to trivially make use of input-side sparsity, since all-zero features can be safely skipped even for zero-padded layers. Because all convolutions can exploit both input- and output-side sparsity, the speed-up gained from pruning is quadratic with respect to the pruning ratio. For instance, dynamically pruning half of the channels in all layers gives rise to a dynamic CNN that uses approximately $\frac{1}{4}$ of the original MACs.

Theoretically, FBS does not introduce the reparameterization discussed in Section 3.2. By batch normalizing the convolution output, the convolution kernel $\boldsymbol{\theta}_l$ is invariant to scaling. Computationally, it is more efficient to train. Many alternative methods use non-differentiable $\pi$ functions that produce on/off decisions. In general, DNNs with these policy functions are incompatible with SGD, and resort to reinforcement learning for training. In contrast, (6) allows end-to-end training, as wta is a piecewise differentiable and continuous function like ReLU. Srivastava et al. (2015) suggested that in general, a network is easier and faster to train for complex tasks and less prone to catastrophic forgetting, if it uses functions such as wta that promote local competition between many subnetworks.

## 3.4 LEARNING TO PREDICT CHANNEL SALIENCIES

This section explains the design of the channel saliency predictor $g_l(\mathbf{x}_{l-1})$. To avoid significant computational cost in $g_l$, we subsample $\mathbf{x}_{l-1}$ by reducing the spatial dimensions of each channel to a scalar using the following function $\mathsf{ss} : \mathbb{R}^{C \times H \times W} \to \mathbb{R}^C$:

$$\mathsf{ss}\left(\mathbf{x}_{l-1}\right) = \frac{1}{HW} \left[ \mathsf{s}\left(\mathbf{x}_{l-1}^{[1]}\right) \; \mathsf{s}\left(\mathbf{x}_{l-1}^{[2]}\right) \; \cdots \; \mathsf{s}\left(\mathbf{x}_{l-1}^{[C]}\right) \right], \tag{7}$$

where $\mathsf{s}\left(\mathbf{x}_{l-1}^{[c]}\right)$ reduces the $c^{\text{th}}$ channel of $\mathbf{z}$ to a scalar using, for instance, the $\ell^1$-norm $\|\mathbf{x}_{l-1}^{[c]}\|_1$, $\ell^2$-norm, $\ell^\infty$-norm, or the variance of $\mathbf{x}_{l-1}^{[c]}$. The results in Section 4 use the $\ell^1$-norm by default, which is equivalent to global average pooling for the ReLU activated $\mathbf{x}_{l-1}$. We then design $g_l(\mathbf{x}_{l-1})$ to predict channel saliencies with a fully connected layer following the subsampled activations $\mathsf{ss}\left(\mathbf{x}_{l-1}\right)$, where $\boldsymbol{\phi}_l \in \mathbb{R}^{C^l \times C^{l-1}}$ is the weight tensor of the layer:

$$g_l\left(\mathbf{x}_{l-1}\right) = \left(\mathsf{ss}\left(\mathbf{x}_{l-1}\right) \boldsymbol{\phi}_l + \boldsymbol{\rho}_l\right)_+. \tag{8}$$

We generally initialize $\boldsymbol{\rho}_l$ with 1 and apply He et al. (2015)'s initialization to $\boldsymbol{\phi}_l$. Similar to how Liu et al. (2017) and Ye et al. (2018) induced sparsity in the BN scaling factors, we regularize all layers with the Lasso on $g_l(\mathbf{x}_{l-1})$: $\lambda \sum_{l=1}^{L} \mathbb{E}_{\mathbf{x}}\left[\|g_l(\mathbf{x}_{l-1})\|_1\right]$ in the total loss, where $\lambda = 10^{-8}$ in our experiments.

## 4 EXPERIMENTS

We ran extensive experiments on CIFAR-10 (Krizhevsky et al., 2014) and the ImageNet ILSVRC2012 (Deng et al., 2009), two popular image classification datasets. We use M-CifarNet (Zhao et al., 2018), a custom 8-layer CNN for CIFAR-10 (see Appendix A for its structure), using only 1.3 M parameters with 91.37% and 99.67% top-1 and top-5 accuracies respectively. M-CifarNet is much smaller than a VGG-16 on CIFAR-10 (Liu et al., 2017), which uses 20 M parameters and only 2.29% more accurate. Because of its compactness, our CNN is more challenging to accelerate. By faithfully reimplementing *Network Slimming* (NS) (Liu et al., 2017), we closely compare FBS with NS under various speedup constraints. For ILSVRC2012, we augment two popular CNN variants, ResNet-18 (He et al., 2016) and VGG-16 (Simonyan & Zisserman, 2015), and provide detailed accuracy/MACs trade-off comparison against recent structured pruning and dynamic execution methods.

Our method begins by first replacing all convolutional layer computations with (5), and initializing the new convolutional kernels with previous parameters. Initially, we do not suppress any channel computations by using density $d = 1$ in (6) and fine-tune the resulting network. For fair comparison against NS, we then follow Liu et al. (2017) by iteratively decrementing the overall density $d$ of the network by 10% in each step, and thus gradually

using fewer channels to sweep the accuracy/performance trade-off. The difference is that NS prunes channels by ranking globally, while FBS prunes around $1 - d$ of each layer.

## 4.1 CIFAR-10

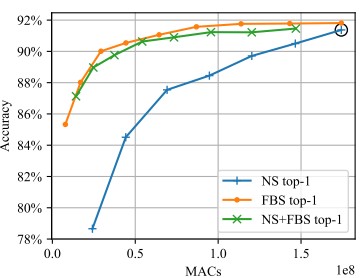

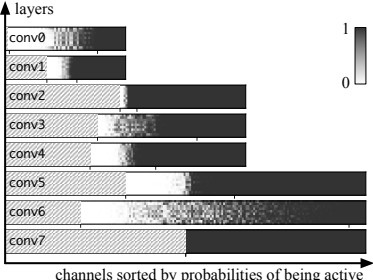

(a) M-CifarNet accuracy/MACs trade-off      (b) Channel skipping probabilites

Figure 3: Experimental results on M-CifarNet. We compare in (a) the accuracy/MACs trade-off between FBS, NS and FBS+NS. The baseline is emphasized by the circle ◯. The heat map in (b) reveals the individual probability of skipping a channel for each channel ($x$-axis), when an image of a category ($y$-axis) is shown to the network with $d = 1$.

By respectively applying NS and FBS to our CIFAR-10 classifier and incrementally increasing sparsity, we produce the trade-off relationships between number of operations (measured in MACs) and the classification accuracy as shown in Figure 3a. FBS clearly surpasses NS in its ability to retain the task accuracy under an increasingly stringent computational budget. Besides comparing FBS against NS, we are interested in combining both methods, which demonstrates the effectiveness of FBS if the model is already less redundant, *i.e.* it cannot be pruned further using NS without degrading the accuracy by more than 1%. The composite method (NS+FBS) is shown to successfully regain most of the lost accuracy due to NS, producing a trade-off curve closely matching FBS. It is notable that under the same 90.50% accuracy constraints, FBS, NS+FBS, and NS respectively achieve 3.93×, 3.22×, and 1.19× speed-up ratios. Conversely for a 2× speed-up target, they respectively produce models with accuracies not lower than 91.55%, 90.90% and 87.54%.

Figure 3b demonstrates that our FBS can effectively learn to amplify and suppress channels when dealing with different input images. The 8 heat maps respectively represent the channel skipping probabilities of the 8 convolutional layers. The brightness of the pixel at location $(x, y)$ denotes the probability of skipping the $x^{\text{th}}$ channel when looking at an image of the $y^{\text{th}}$ category. The heat maps verify our belief that the auxiliary network learned to predict which channels specialize to which features, as channels may have drastically distinct probabilites of being used for images of different categories. The model here is a M-CifarNet using FBS with $d = 0.5$, which has a top-1 accuracy of 90.59% (top-5 99.65%). Moreover, channels in the heat maps are sorted so the channels that are on average least frequently evaluated are placed on the left, and channels shaded in stripes are never evaluated. The network in Figure 3b is not only approximately 4× faster than the original, by removing the unused channels, we also reduce the number of weights by 2.37×. This reveals that FBS naturally subsumes channel pruning strategies such as NS, as we can simply prune away channels that are skipped regardless of the input. It is notable that even though we specified a universal density $d$, FBS learned to adjust its dynamicity across all layers, and prune different ratios of channels from the convolutional layers.

## 4.2 ImageNet ILSVRC2012 Classification

By applying FBS and NS respectively to ResNet-18, we saw that the ILSVRC2012 validation accuracy of FBS consistently outperforms NS under different speed-up constraints (see Appendix B for the implementation details and trade-off curves). For instance, at $d = 0.7$, it utilizes only 1.12 G MACs (1.62× fewer) to achieve a top-1 error rate of 31.54%, while NS requires 1.51 G MACs (1.21× fewer) for a similar error rate of 31.70%. When compared

across recent dynamic execution methods examined in Table 1, FBS demonstrates simultaneously the highest possible speed-up and the lowest error rates. It is notable that the baseline accuracies for FBS refer to a network that has been augmented with the auxiliary layers featuring FBS but suppress no channels, *i.e.* $d = 1$. We found that this method brings immediate accuracy improvements, an increase of 1.73% in top-1 and 0.46% in top-5 accuracies, to the baseline network, which is in line with our observation on M-CifarNet.

In Table 2, we compare different structured pruning and dynamic execution methods to FBS for VGG-16 (see Appendix B for the setup). At a speed-up of $3.01\times$, FBS shows a minimal increase of 0.44% and 0.04% in top-1 and top-5 errors respectively. At $5.23\times$ speed-up, it only degrades the top-1 error by 1.08% and the top-5 by 0.59%.

Not only does FBS use much fewer MACs, it also demonstrates significant reductions in bandwidth and memory requirements. In Table 3, we observe a large reduction in the number of memory accesses in single image inference as we simply do not access suppressed weights and activations. Because these memory operations are often costly DRAM accesses, minimizing them leads to power-savings. Table 3 further reveals that in diverse application scenarios such as low-end and cloud environments, the peak memory usages by the optimized models are much smaller than the originals, which in general improves cache utilization.

| Method | Dynamic | Baseline | | Accelerated | | MAC |
| | | Top-1 | Top-5 | Top-1 | Top-5 | saving |
|---|---|---|---|---|---|---|
| *Soft Filter Pruning* (He et al., 2018a) | | 29.72 | 10.37 | 32.90 | 12.22 | 1.72× |
| *Network Slimming* (Liu et al. (2017), our implementation) | | 31.02 | 11.32 | 32.79 | 12.61 | 1.39× |
| *Discrimination-aware Channel Pruning* (Zhuang et al., 2018) | | 30.36 | 11.02 | 32.65 | 12.40 | 1.89× |
| *Low-cost Collaborative Layers* (Dong et al., 2017) | ✓ | 30.02 | 10.76 | 33.67 | 13.06 | 1.53× |
| *Channel Gating Neural Networks* (Hua et al., 2018) | ✓ | 30.98 | 11.16 | 32.60 | 12.19 | 1.61× |
| *Feature Boosting and Suppression* (FBS) | ✓ | **29.29** | **10.32** | **31.83** | **11.78** | **1.98×** |

Table 1: Comparisons of error rates of the baseline and accelerated ResNet-18 models.

| Method | Dynamic | Δ top-5 errors (%) | | |
| | | 3× | 4× | 5× |
|---|---|---|---|---|
| *Filter Pruning* (Li et al. (2017), reproduced by He et al. (2017)) | | — | 8.6 | 14.6 |
| *Perforated CNNs* (Figurnov et al., 2016) | | 3.7 | 5.5 | — |
| *Network Slimming* (Liu et al. (2017), our implementation) | | 1.37 | 3.26 | 5.18 |
| *Runtime Neural Pruning* (Lin et al., 2017) | ✓ | 2.32 | 3.23 | 3.58 |
| *Channel Pruning* (He et al., 2017) | | 0.0 | 1.0 | 1.7 |
| *AutoML for Model Compression* (He et al., 2018b) | | — | — | 1.4 |
| *ThiNet-Conv* (Luo et al., 2017) | | 0.37 | — | — |
| *Feature Boosting and Suppression* (FBS) | ✓ | 0.04 | **0.52** | **0.59** |

Table 2: Comparisons of top-5 error rate increases for VGG-16 on ILSVRC2012 validation set under $3\times$, $4\times$ and $5\times$ speed-up constraints. The baseline has a 10.1% top-5 error rate. Results from He et al. (2017) only show numbers with one digit after the decimal point.

| Model | Total Memory Accesses | | Peak Memory Usage | |
| | Weights | Activations | Edge (1 image) | Cloud (128 images) |
|---|---|---|---|---|
| *VGG-16* | 56.2 MB | 86.5 MB | 24.6 MB | 3.09 GB |
| *VGG-16* 3× | 23.9 MB (2.35×) | 40.8 MB (2.12×) | 9.97 MB (2.47×) | 1.24 GB (2.47×) |
| *ResNet-18* | 44.6 MB | 17.8 MB | 9.19 MB | 0.47 GB |
| *ResNet-18* 2× | 20.5 MB (2.18×) | 12.3 MB (1.45×) | 4.68 MB (1.96×) | 0.31 GB (1.49×) |

Table 3: Comparisons of the memory accesses and peak memory usage of the ILSVRC2012 classifiers with FBS respectively under $3\times$ and $2\times$ inference speed-ups. The **Weights** and **Activations** columns respectively show the total amount of weight and activation accesses required by all convolutions for a single image inference. The **Peak Memory Usage** columns show the peak memory usages with different batch sizes.

## 5 CONCLUSION

In summary, we proposed feature boosting and suppression that helps CNNs to achieve significant reductions in the compute required while maintaining high accuracies. FBS fully

preserves the capabilities of CNNs and predictively boosts important channels to help the accelerated models retain high accuracies. We demonstrated that FBS achieves around $2\times$ and $5\times$ savings in computation respectively on ResNet-18 and VGG-16 within 0.6% loss of top-5 accuracy. Under the same performance constraints, the accuracy gained by FBS surpasses all recent structured pruning and dynamic execution methods examined in this paper. In addition, it can serve as an off-the-shelf technique for accelerating many popular CNN networks and the fine-tuning process is unified in the traditional SGD which requires no algorithmic changes in training. Finally, the implementation of FBS and the optimized networks are fully open source and released to the public[1].

## ACKNOWLEDGEMENTS

This work is supported in part by the National Key R&D Program of China (No. 2018YFB1004804), the National Natural Science Foundation of China (No. 61806192). We thank EPSRC for providing Yiren Zhao his doctoral scholarship.

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

## A    Details of M-CifarNet on CIFAR-10

For the CIFAR-10 classification task, we use M-CifarNet, a custom designed CNN, with less than 1.30 M parameters and takes 174 M MACs to perform inference for a 32-by-32 RGB image. The architecture is illustrated in Table 4, where all convolutional layers use $3 \times 3$ kernels, the **Shape** column shows the shapes of each layer's features, and `pool7` is a global average pooling layer.

We trained M-CifarNet (see Appendix A) with a 0.01 learning rate and a 256 batch size. We reduced the learning rate by a factor of $10\times$ for every 100 epochs. To compare FBS against NS fairly, every model with a new target MACs budget were consecutively initialized with the previous model, and trained for a maximum of 300 epochs, which is enough for all models to converge to the best obtainable accuracies. For NS, we follow Liu et al. (2017) and start training with an $\ell^1$-norm sparsity regularization weighted by $10^{-5}$ on the BN scaling factors. We then prune at 150 epochs and fine-tune the resulting network without the sparsity regularization.

We additionally employed image augmentation procedures from Krizhevsky et al. (2012) to preprocess each training example. Each CIFAR-10 example was randomly horizontal flipped and slightly perturbed in the brightness, saturation and hue.

Table 4 additionally provides further comparisons of layer-wise compute costs between FBS, NS, and the composition of the two methods (NS+FBS). It is notable that the FBS column has two different output channel counts, where the former is the number of computed channels for each inference, and the latter is the number of channels remaining in the layer after removing the unused channels.

| Layer | Shape | Number of MACs (Output Channels) | | | |
|-------|-------|----------|-----|-----|--------|
| | | Original | NS | FBS | NS+FBS |
| `conv0` | $30 \times 30$ | 1.5 M (64) | 1.3 M (52) | 893 k (32/62) | 860 k (32) |
| `conv1` | $30 \times 30$ | 33.2 M (64) | 27.0 M (64) | 8.4 M (32/42) | 10.2 M (39) |
| `conv2` | $15 \times 15$ | 16.6 M (128) | 15.9 M (123) | 4.2 M (64/67) | 5.9 M (74) |
| `conv3` | $15 \times 15$ | 33.2 M (128) | 31.9 M (128) | 8.3 M (64/79) | 11.6 M (77) |
| `conv4` | $15 \times 15$ | 33.2 M (128) | 33.1 M (128) | 8.3 M (64/83) | 12.1 M (77) |
| `conv5` | $8 \times 8$ | 14.1 M (192) | 13.4 M (182) | 3.6 M (96/128) | 4.9 M (110) |
| `conv6` | $8 \times 8$ | 21.2 M (192) | 11.6 M (111) | 5.4 M (96/152) | 4.3 M (67) |
| `conv7` | $8 \times 8$ | 21.2 M (192) | 12.3 M (192) | 5.4 M (96/96) | 4.5 M (116) |
| `pool7` | $1 \times 1$ | | | | |
| `fc` | $1 \times 1$ | 1.9 k (10) | 1.9 k (10) | 960 (10) | 1.1 k (10) |
| Total | | 174.3 M | 146.5 M | 44.3 M | 54.2 M |
| Saving | | - | $1.19\times$ | $3.93\times$ | $3.21\times$ |

Table 4: The network structure of M-CifarNet for CIFAR-10 classification. In addition, we provide a detailed per-layer MACs comparison between FBS, NS, and the composition of them (NS+FBS). We minimize the models generated by the three methods while maintaining a classification accuracy of at least 90.5%.

Figure 4 shows how the skipping probabilites heat maps of the convolutional layer `conv4` evolve as we fine-tune FBS-augmented M-CifarNet. The network was trained for 12 epochs, and we saved the model at every epoch. The heat maps are generated with the saved models in sequence, where we apply the same reordering to all heat map channels with the sorted result from the first epoch. It can be observed that as we train the network, the channel skipping probabilites become more pronounced.

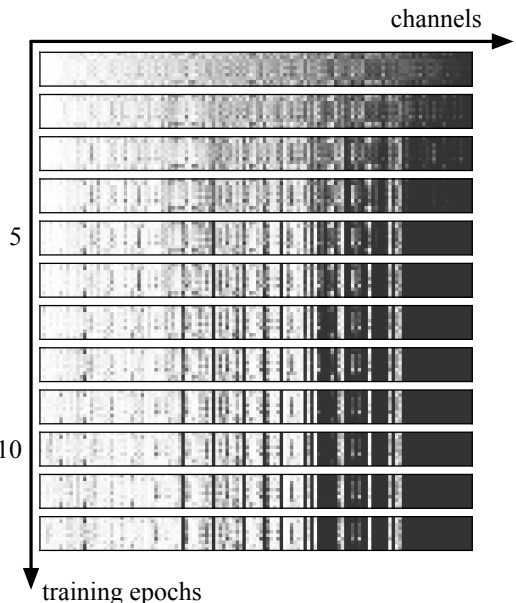

Figure 4: The training history of a convolutional layer `conv4` in M-CifarNet. The history is visualized by the 12 skipping probabilites heat maps, where the heights denote the 10 categories in CIFAR-10, and channels in `conv4` occupy the width.

## B  Details of the ILSVRC2012 classifiers

ILSVRC2012 classifiers, *i.e.* ResNet-18 and VGG-16, were trained with a procedure similar to Appendix A. The difference was that they were trained for a maximum of 35 epochs, the learning rate was decayed for every 20 epochs, and NS models were all pruned at 15 epochs. For image preprocessing, we additionally cropped and stretched/squeezed images randomly following Krizhevsky et al. (2012).

Since VGG-16 is computationally intensive with over 15 G MACs, We first applied NS on VGG-16 to reduce the computational and memory requirements, and ease the training of the FBS-augmented variant. We assigned a 1% budget in top-5 accuracy degradation and compressed the network using NS, which gave us a smaller VGG-16 with 20% of all channels pruned. The resulting network is a lot less redundant, which almost halves the compute requirements, with only 7.90 G MACs remaining. We then apply FBS to the well-compressed network.

Residual networks (He et al., 2016), such as ResNet-18, adopt sequential structure of residual blocks: $\mathbf{x}_b = K\left(\mathbf{x}_{b-1}\right) + F\left(\mathbf{x}_{b-1}\right)$, where $\mathbf{x}_b$ is the output of the $b^{\text{th}}$ block, $K$ is either an identity function or a downsampling convolution, and $F$ consists of a sequence of convolutions. For residual networks, we directly apply FBS to all convolutional layers, with a difference in the way we handle the feature summation. Because the $(b+1)^{\text{th}}$ block receives as input the sum of the two features with sparse channels $K\left(\mathbf{x}_{b-1}\right)$ and $F\left(\mathbf{x}_{b-1}\right)$, a certain channel of this sum is treated as sparse when the same channels in both features are simultaneously sparse.

Figure 5 compares the accuracy/performance trade-off curves between FBS and NS for ResNet-18.

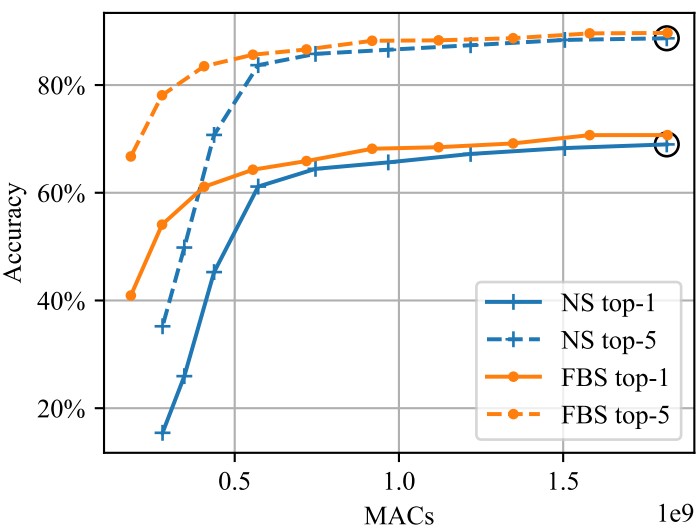

Figure 5: The accuracy/performance trade-off comparison between NS and FBS for ResNet-18 on the ImageNet ILSVRC2012 validation set.

