# OpenReview forum: "Dynamic Channel Pruning: Feature Boosting and Suppression"
_ICLR.cc/2019/Conference_

### Official Review · AnonReviewer3 · 2018-11-03
**Review comments on “Dynamic Channel Pruning: Feature Boosting and Suppression”**

**Rating:** 6
**Confidence:** 4

**Review:**

Summary:

This paper proposed a feature boosting and suppression method for dynamic channel pruning. To be specific, the proposed method firstly predicts the importance of each channel and then use an affine function to amplify/suppress the importance of different channels. However, the idea of dynamic channel pruning is not novel. Moreover, the comparisons in the experiments are quite limited.

My detailed comments are as follows.


Strengths:

1. The motivation for this paper is reasonable and very important.

2. The authors proposed a new method for dynamic channel pruning.

Weaknesses:

1. The idea of dynamic channel pruning is not novel. In my opinion, this paper is only an extension to Network Slimming (Liu et al., 2017). What is the essential difference between the proposed method and Network Slimming?

2. The writing and organization of this paper need to be significantly improved. There are many grammatical errors and this paper should be carefully proof-read.

3. The authors argued that the importance of features is highly input-dependent. This problem is reasonable but the proposed method still cannot handle it. According to Eqn. (7), the prediction of channel saliency relies on a data batch rather than a single data. Given different inputs in a batch, the selected channels should be different for each input rather than a general one for the whole batch. Please comment on this issue.

4. The proposed method does not remove any channels from the original model. As a result, both the memory and the computational cost will not be reduced. It is confusing why the proposed method can yield a significant speed-up in the experiments.

5. The authors only evaluate the proposed method on shallow models, e.g., VGG and ResNet18. What about the deeper model like ResNet50 on ImageNet?

6. It is very confusing why the authors only reported top-5 error of VGG. The results of top-1 error for VGG should be compared in the experiments.

7. Several state-of-the-art channel pruning methods should be considered as the baselines, such as ThiNet (Luo et al., 2017), Channel pruning (He et al., 2017) and DCP (Zhuang et al., 2018)
[1] Channel pruning for accelerating very deep neural networks. CVPR 2017.
[2] Thinet: A filter level pruning method for deep neural network compression. CVPR 2017.
[3] Discrimination-aware Channel Pruning for Deep Neural Networks. NIPS 2018.

---

> ### Author Response · Authors · 2018-11-06
> **Reply to Reviewer 3 (2/2)**
>
> 3. "The authors argued that the importance of features is highly input-dependent. This problem is reasonable but the proposed method still cannot handle it.
> According to Eqn. (7), the prediction of channel saliency relies on a data batch rather than a single data. Given different inputs in a batch, the selected channels should be different for each input rather than a general one for the whole batch. Please comment on this issue."
>
> The prediction of channel saliency *does not* rely on a batch of data. In equation (7), x_(l-1) is the output of the (l-1)-th layer, which comprises of C_(l-1) features, each feature has the spatial dimensions H_(l-1) * W_(l-1), as defined in Section 3.1. Throughout this paper, x_l for all layers is a single input image, which consists of multiple channels. Equation (7) reduces each channel in an image to a scalar, which is then used to predict the output channel saliencies in equation (8). Although this process is identical for each input image, each evaluation of equation (8) may produce drastically different predicted channel saliencies dependent on the input image.
>
> We would like to update this section to remove any sources of ambiguity, would it be possible for you to describe how our intended meaning was lost?
>
> 4. "The proposed method does not remove any channels from the original model. As a result, both the memory and the computational cost will not be reduced. It is confusing why the proposed method can yield a significant speed-up in the experiments.”
>
> It is hopefully clear from previous comments that this is not the case.
>
> Typically, convolutional layers are stacked to form a sequential convolutional network. Prior to computing the costly convolution, FBS uses the input (or the output from the previous layer) to predict the saliencies of output channels of the costly convolution. If an output channel is predicted to have a zero saliency, the evaluation of this output channel can be entirely skipped, as the entire output channel is predicted to contain only zero entries.
>
> In addition, each convolutional layer takes as its input the output of the previous layer. This input can have channel-wise sparsity (channels consisting of only zero entries), if the previous layer is a convolutional layer. It is clear that these inactive input channels can always be skipped when computing the convolution.
>
> The input- and output-side sparsities therefore doubly accelerate the expensive convolution and thus achieve a huge reduction in compute. Such reduction in computation is also seen in [2], as it shares the same goal but uses an entirely different method.
>
> 5. "The authors only evaluate the proposed method on shallow models, e.g., VGG and ResNet18. What about the deeper model like ResNet50 on ImageNet?"
>
> The method we propose is a per-layer method, which should not make a difference when targeting deeper models. Unlike NS, we do not rank channel importance globally to produce pruning decisions. We are working on generating results on deeper models, but this might be limited by the amount of time available.
>
> 6. "It is very confusing why the authors only reported top-5 error of VGG. The results of top-1 error for VGG should be compared in the experiments."
>
> We will update Table 2 to include top-1 errors. However, some works we compare to, e.g. He et al.'s channel pruning [4], may have missing top-1 errors as they were not reported.
>
> 7. "Several state-of-the-art channel pruning methods should be considered as the baselines, such as ThiNet (Luo et al., 2017), Channel pruning (He et al., 2017) and DCP (Zhuang et al., 2018)."
>
> Thank you for pointing out these works. These are all static techniques. We will be including them in our comparisons. In addition, it should be noted that Channel pruning [4] is already in our comparison of Table 2.
>
>
> We thank the reviewer for providing this review.
>
> We are in the process of updating this paper, and will notify you by comment of the new revision and its changes.
>
> [1]: Squeeze-and-Excitation Networks, CVPR 2018, https://arxiv.org/abs/1709.01507
> [2]: Runtime Neural Pruning, NIPS 2017, https://papers.nips.cc/paper/6813-runtime-neural-pruning
> [3]: Conditional Computation in Neural Networks for Faster Models, ICLR 2016, https://arxiv.org/abs/1511.06297
> [4]: Channel pruning for accelerating very deep neural networks, ICCV 2017, https://arxiv.org/abs/1707.06168

---

> ### Author Response · Authors · 2018-11-06
> **Reply to Reviewer 3 (1/2)**
>
> Thanks for your review.
>
> We would like to clarify some points to avoid misunderstandings.
>
> Our paper proposes a method called Feature Boosting and Suppression (FBS). FBS adds small auxiliary layers on top of each existing convolution. These auxiliary layers have trainable parameters that are optimized using SGD and control whether individual channels are evaluated at run-time or not. Using this conditional execution, the overall computation required is reduced significantly. Furthermore, the output of the auxiliary layers is used to scale each channel output. Channel saliencies are computed by the auxiliary layers on a per input basis. FBS utilizes sparse input channels (from the previous dynamically pruned convolutional layer) to predict which channels to skip in the output channels, so that we have large reduction in computations, as we exploit both input- and output-side sparsities.
>
> The weaknesses identified by the reviewer (1,3 and 4) do not hold for the approach described above. We will address each of these comments in turn.
>
> Introductory statement:
> "firstly predicts the importance of each channel and then use an affine function to amplify/suppress the importance of different channels"
>
> This statement is not true. To clarify, the amplification of channels is dependent on the input (Equation 5), whereas the suppression process effectively performs important channel selection (Equation 6). Both yield strictly non-affine transformations on the batch normalized channel output.
>
> 1. "The idea of dynamic channel pruning is not novel. In my opinion, this paper is only an extension to Network Slimming (Liu et al., 2017).
> What is the essential difference between the proposed method and Network Slimming?"
>
> The Network Slimming (NS) procedure is applied statically and only prunes channels away. Our technique is applied at run-time and is input dependent. We prune channels away and boost important channels at run-time.
>
> We consider our method, FBS, to be very different from Network Slimming. For each input image during inference, FBS predicts the relative importance of each channel, and selectively evaluates a subset of output channels that are important for the subsequent layer, given the activation of the previous layer. Different input images would therefore activate drastically different execution paths in the model.
>
> Figure 3b corroborates this observation, as the heat maps show that many channels demonstrate high varying probabilities of being suppressed when being shown images of different categories. Our work is more related to runtime neural pruning [2] and conditional computation [3], where channels are dynamically selected for evaluation in each convolution, yet [2], [3] and FBS use very different methods to achieve this goal. In contrast, NS does not employ dynamic execution, as the pruned channels are *permanently removed* from the model, resulting in a network structure that remains static for all inputs where some capabilities will be permanently lost.
>
> In addition, FBS preemptively steers feature attention: as FBS not only uses the saliency metrics to predicatively prune unimportant channels at run-time, it further amplifies important channels. The non-linearity added to the network is conceptually similar to Squeeze-and-Excitation (SE) [1], as FBS captures inter-dependencies among input channels and adaptively recalibrates output features in a channel-wise fashion. Even without pruning, FBS can improve the baseline accuracies of CIFAR-10 and ImageNet models (Section 4.2), which is absent from static/dynamic channel pruning methods including NS, RNP, [4] and others.
>
> Because of the above differences, FBS can achieve a much improved accuracy/compute trade-off when compared to other channel pruning methods.
>
> 2. "The writing and organization of this paper need to be significantly improved. There are many grammatical errors and this paper should be carefully proof-read."
>
> We will complete another round of polishing to address any shortcomings. Could you suggest how/where the organization of the paper could be improved?

---

> ### Author Response · Authors · 2018-11-21
> **We've updated a new revision of our submission.**
>
> I hope this addresses weaknesses 2, 6 and 7 identified by your comments. We additionally included more comparisons against other works in Tables 1 and 2.

---

> > ### Comment · AnonReviewer3 · 2018-12-06
> > **further comments**
> >
> > In the revision, the authors have made significant improvement over the original submission. I also appreciate that my main concerns regarding the original submission have been addressed.

---

### Official Review · AnonReviewer2 · 2018-11-06
**feature suppression to speed up training CNN**

**Rating:** 7
**Confidence:** 4

**Review:**

This manuscript presents a nice method that can dynamically prune some channels in a CNN network to speed up the training. The main strength of the proposed method is to determine which channels to be suppressed based upon each data sample without incurring too much computational burden or too much memory consumption.  The good thing is that the proposed pruning strategy does not result in a big performance decrease. Overall, this is a nicely written paper and may be empirically useful for training a very large CNN. Nevertheless, the authors did not present a real-world application in which it is important to speed up by 2 or 3 times at a small cost, so it is hard to judge the real impact of the proposed method.

---

> ### Author Response · Authors · 2018-11-21
> **Thank you for your comments.**
>
> > "the authors did not present a real-world application in
> > which it is important to speed up by 2 or 3 times at a small
> > cost, so it is hard to judge the real
> > impact of the proposed method."
>
> Of course, all real systems are constrained by power and memory bandwidth. The proposed scheme offers very significant savings (2-3X in both compute and memory bandwidth) that would be beneficial in almost all scenarios, either to reduce power, increase performance or trade for better accuracy.
>
> Additionally, we would like to point out that FBS works as an technique to accelerate network inference.  Although it is entirely feasible to use it to accelerate training, we have not conducted relevant experiments.

---

### Official Review · AnonReviewer4 · 2018-11-12
**Review for "Dynamic Channel Pruning: Feature Boosting and Suppression"**

**Rating:** 6
**Confidence:** 3

**Review:**

The authors propose a dynamic inference technique for accelerating neural network prediction with minimal accuracy loss.  The technique prunes channels in an input-dependent way through the addition of auxiliary channel saliency prediction+pruning connections.

Pros:
- The paper is well-written and clearly explains the technique, and Figure 1 nicely summarizes the weakness of static channel pruning
- The technique itself is simple and memory-efficient
- The performance decrease is small

Cons:
- There is no clear motivation for the setting (keeping model accuracy while increasing inference speed by 2x or 5x)
- In contrast to methods that prune weights, the model size is not reduced, decreasing the utility in many settings where faster inference and smaller models are desired (e.g. mobile, real-time)
- The experiments are limited to classification and fairly dated architectures (VGG16, ResNet-18)

Overall, the method is nicely explained but the motivation is not clear.  Provided that speeding up inference without reducing the size of the model is desirable, this paper gives a good technique for preserving accuracy.

---

> ### Author Response · Authors · 2018-11-21
> **Reply to Reviewer 4**
>
> Thank you for your comments.
>
> 1. Re. motivation, to clarify we do increase performance as you state (2--5x) but in addition also make significant savings in terms of compute and memory bandwidth. These savings would be beneficial in almost all scenarios, either to reduce power, increase performance or trade for better accuracy. We have clarified this in our introduction.
>
> 2. I think there is some misunderstanding here. By dynamically gating computation, FBS reduces both compute and memory requirements. We simply don't load/store the weights/activations for the suppressed channels. The newly added Table 3 quantifies these savings.
>
> 3. We are working on generating data for newer models, but this might be limited by the amount of time available.

---

### Official Review · AnonReviewer5 · 2018-11-22
**review comments on “Dynamic Channel Pruning: Feature Boosting and Suppression”**

**Rating:** 7
**Confidence:** 5

**Review:**

This paper propose a channel pruning method for dynamically selecting channels during testing. The analysis has shown that some channels are not always active.

Pros:
- The results on ImageNet are promising. FBS achieves state-of-the-art results on VGG-16 and ResNet-18.
- The method is simple yet effective.
- The paper is clear and easy to follow.

Cons:
- Lack of experiments on mobile networks like shufflenets and mobilenets
- Missing citations of some state-of-the-art methods [1] [2].
- The speed-up ratios on GPU or CPU are not demonstrated. The dynamic design of Dong et al., 2017 did not achieve good GPU speedup.
- Some small typos.

[1] Amc: Automl for model compression and acceleration on mobile devices
[2] Netadapt: Platform-aware neural network adaptation for mobile applications

---

> ### Author Response · Authors · 2018-12-10
> **Thank you for your comments.**
>
> We would like to thank the reviewer for the positive comments.
> A comparison to AMC [1] is included in Table 2,  it is difficult for us to compare to Netadapt [2] since the networks considered are different.
> We would like to point out that Dong et al. [3] considered spatial dynamic execution, which eliminates computations at a finer granularity and is thus harder to accelerate compared to our channel-wise dynamic execution. On a CPU, we recently found that a single layer using FBS can increase inference speed by 3.91x, given a theoretical speedup of 3.98x.
>
> [3] More is Less: A More Complicated Network with Less Inference Complexity, CVPR 2017, https://arxiv.org/pdf/1703.08651.pdf

---

### Comment · Area_Chair1 · 2018-12-09
**misleading to report "FLOP reduction" as "speedup"**

It's misleading to the community to report "FLOP reduction" as "speedup". FLOP reduction doesn't translate to speedup on hardware. If the authors wants to report the speedup, please report the wall-clock time support the below claim: "FBS cam accelerate VGG-16 by 5× and improve the speed of ResNet-18 by 2×"

---

> ### Author Response · Authors · 2018-12-10
> **Thank you for your comments.**
>
> Thanks for the correction, we will change this to a more precise statement: "FBS can reduce the FLOPs of VGG-16 by 5x and ResNet-18 by 2x".
>
> We tested on CPU one layer of VGG-16 (the 2nd convolution layer) with FBS using the new Pytorch 1.0 (JIT enabled), and achieved 3.91x speedup in wall-clock time when the FBS density is set to 0.5 (which yields a theoretical speedup of 3.98x). FBS achieves a wall-clock time of 12.780ms and the original convolution takes 49.942ms. This minor overhead is mostly due to the excessive data movements to dynamically gather a subset of weight parameters that cannot be eliminated because of the API limitations. We will put the details of this wall-clock time test in Appendix with open source code if accepted.
>
> Given that relatively large blocks of compute can be omitted, it is realistic to suggest that in this case, FLOP reduction will translate into wall-clock time savings. We foresee no particular problems in doing this but existing hardware and tool-chains may currently prevent the necessary optimisations. We would certainly agree that if the optimisations focused on eliminating computations at a finer granularity that actual gains may be difficult to obtain.

---

> > ### Comment · Area_Chair1 · 2018-12-12
> > **don't cherry pick**
> >
> > Please report the wall-clock time running the *whole network* on VGG-16 and ResNet-18, rather than cherry picking a specific layer to show speedup. The last column of Table 1 is not "speedup", but "FLOP reduction".

---

> > > ### Author Response · Authors · 2018-12-18
> > > **Network performance results as requested.**
> > >
> > > We tested VGG-16 and ResNet-18 with FBS against their respective baselines, the experiments were repeated 1000 times and we recorded the average wall-clock time results for each model.
> > >
> > > The VGG-16 baseline observed on average 520.80 ms for each inference.  FBS was applied to VGG-16 and reduced the amount of computation by a factor of 3.01x.  Inference now took 175.13ms, thus achieving a speedup of 2.97x (in terms of wall-clock time).  Similarly, a model with 4.00x computation reduction took 142.17 ms, which translates to a 3.66x actual speedup.  This means that the overhead of our PyTorch implementation is less than 10%.
> > >
> > > A line-by-line profiling of our implementation revealed that the overhead of the extra computations introduced by FBS in convolutional layers are fairly minimal (we have annotated the percentage of execution time of each component here: https://imgur.com/YVQormC ).  We found that the excessive data movements we mentioned earlier contribute to the majority of the observed overhead, while actual computations introduced by FBS amount to only 3.0% of the total time required to compute the layers.  As we have suggested, the data movements are entirely redundant due to API limitations.
> > >
> > > Our FBS-based ResNet-18 provided a 1.98x reduction in the amount of computation, which took 63.73 ms for each inference, while the baseline required 101.82 ms, thus achieving a 1.60x real performance gain.  We found that in addition to the overhead introduced by the FBS implementation above, the add operations for residuals cannot be accelerated in PyTorch for channel-wise sparse activations, and incur excessive copy operations as a result of the API limitations.  Even with these limitations, the real speedup provided by FBS surpasses/matches the actual speedups of all other works compared in Table 1:
> > > ------------------------------------------------------------------ -------------- ------------- ------
> > > Method                                                                                  Top-5 error  Theoretical  Real
> > > ------------------------------------------------------------------ -------------- ------------- ------
> > > Soft Filter Pruning (He et al., 2018)                                            12.22%           1.72x  1.38x
> > > Discrimination-aware Channel Pruning (Zhuang et al., 2018)    12.40%           1.85x  1.60x
> > > Low-cost Collaborative Layers (Dong et al., 2017)                    13.06%           1.53x  1.25x
> > > Feature Boosting and Suppression (this work)                          11.78%           1.98x  1.60x
> > > ------------------------------------------------------------------ -------------- ------------- ------
> > >
> > > We hope this answers your concern regarding the actual performance gains.

---

### Public Comment · ~Nikolaos_Fragkoulis1 · 2019-04-18
**Similar Work**

There is a similar (if not identical) work already published here (https://arxiv.org/abs/1701.05221). Please at least consider adding it to the references

---

### Meta-Review · Area_Chair1 · 2018-12-17
**borderline**

**Confidence:** 3
**Recommendation:** Accept (Poster)

**Metareview:**

The authors propose a dynamic inference technique for accelerating neural network prediction with minimal accuracy loss. The method are simple and effective. The paper is clear and easy to follow. However, the real speedup on CPU/GPU is not demonstrated beyond the theoretical FLOPs reduction. Reviewers are also concerned that the idea of dynamic channel pruning is not novel. The evaluation is on fairly old networks.